Review

Subject Area:
cellular biology/genetics/molecular biology

Keywords:
cyclin-dependent kinases, CDK5, cancer, development, non-neuronal functions

Author for correspondence:
Piotr Sicinski
e-mail: peter_sicinski@dfci.harvard.edu

# A kinase of many talents: non-neuronal functions of CDK5 in development and disease

Samanta Sharma[1,2] and Piotr Sicinski[1,2]

[1]Department of Cancer Biology, Dana-Farber Cancer Institute, Boston, MA, USA
[2]Department of Genetics, Blavatnik Institute, Harvard Medical School, Boston, MA 02215, USA

 SS, 0000-0001-9988-8684

The cyclin-dependent kinase 5 (CDK5) represents an unusual member of the family of cyclin-dependent kinases, which is activated upon binding to non-cyclin p35 and p39 proteins. The role of CDK5 in the nervous system has been very well established. In addition, there is growing evidence that CDK5 is also active in non-neuronal tissues, where it has been postulated to affect a variety of functions such as the immune response, angiogenesis, myogenesis, melanogenesis and regulation of insulin levels. Moreover, high levels of CDK5 have been observed in different tumour types, and CDK5 was proposed to play various roles in the tumorigenic process. In this review, we discuss these various CDK5 functions in normal physiology and disease, and highlight the therapeutic potential of targeting CDK5.

## 1. Introduction

Cyclin-dependent kinases (CDKs) constitute a family of serine/threonine kinases that regulate progression through different phases of the eukaryotic cell cycle [1]. The function of CDKs in cell cycle progression and transcriptional regulation has been very well established. In this review, we will focus on a distinct CDK, namely CDK5. CDK5 was discovered by purification from bovine brain [2]. At the same time, several groups independently identified CDK5 based on its sequence homology to yeast cdc2, mouse Cdk1 and human CDK2 and referred to it by multiple names such as neuronal CDC2-like kinase (NCLK), brain proline-directed proline kinase (BPDK) or PSSALRE [2–6].

Human CDK5 shares 58% and 62% sequence homology with human CDK1 and CDK2, respectively [7]. Like other CDKs, CDK5 is a proline-directed kinase that phosphorylates consensus (S/T)PX(K/H/R) sequence [8,9]. However, CDK5 is in many ways distinct from 'classical' CDKs. First, unlike other CDKs—which are activated by cyclin subunits—the catalytic activity of CDK5 is triggered upon interaction with p35 and p39 proteins, which do not share sequence similarity with cyclins [10,11]. Another major difference is that the classical CDKs require an activating phosphorylation at threonine 160 (carried out by the CDK-activating kinase, CAK) for their catalytic activity. By contrast, CDK5 does not require such phosphorylation, despite the sequence similarity and the presence of serine159 at a position equivalent to Thr160 of CDK2 [11,12]. Indeed, the binding of p35 and p39 to CDK5 alone is sufficient to fully activate CDK5 [11]. Lastly, cell cycle inhibitors p27[Kip1], p57[Kip2] and p21[Cip1] inhibit the kinase activity of cyclin-CDK1 and cyclin-CDK2 complexes, but they have essentially no effect on p35/p39–CDK5 kinase [13,14]. Collectively, these observations established that p35/p39–CDK5 kinase is molecularly distinct from classical cyclin–CDK complexes.

Although CDK5 is expressed ubiquitously, its activity was thought to be restricted to the nervous system. Indeed, CDK5 activators p35 and p39

are mainly expressed in post-mitotic neurons, explaining tissue-specific activity of CDK5 [10,15]. The functions of p35/p39–CDK5 kinase in the nervous system have been extensively studied and described in several reviews [7,16–18]; hence, this subject will not be covered here. Briefly, CDK5 was shown to control virtually all aspects of neuronal physiology (through phosphorylation of a wide range of neuronal proteins), such as neuronal migration, neurite outgrowth, dendritic arborization, axonal elongation, synapse formation, synaptic plasticity, memory formation and pain signalling [10,19–25]. Mice lacking Cdk5 die perinatally and display a striking inversion of neuronal layering in the cerebral cortex, hippocampus, cerebellum and olfactory bulb, due to a defect in neuronal migration [26]. While mice lacking p35 and p39 are viable, a combined ablation of p35 and p39 phenocopies Cdk5 deficiency and results in a block in neuronal migration [27,28].

Hyper-activation of CDK5 is seen in a number of neurodegenerative diseases where it is believed to play a causative role in the degenerative process. It is assumed that hyperactive CDK5 phosphorylates non-physiological substrates resulting in neuronal pathology. In Alzheimer's disease (AD), neuronal stress leads to calpain-dependent cleavage of p35 into hyper-stable p25 species, leading to hyper-activation of CDK5. Hyperactive CDK5 phosphorylates amyloid precursor protein (APP), tau and neurofilaments, and these events contribute to AD pathogenesis [29,30]. Hyper-activation of CDK5 has also been implicated in the pathogenesis of Parkinson's disease, amyotrophic lateral sclerosis (ALS) and prion-related encephalopathies [13,31–33].

While most studies of CDK5 focused on the nervous system, its non-neuronal functions have been overlooked for a long time. However, there is an increasing evidence that CDK5 may play an important role outside the nervous system and that it may regulate a diverse array of functions, such as cell cycle progression, apoptosis, DNA damage response, metabolism, angiogenesis, myogenesis, immune function, cell migration, invasion and epithelial to mesenchymal transition [22,34]. Here, we review these non-neuronal functions of CDK5 (figures 1a and 2a) and discuss the potential of targeting CDK5 for therapeutic intervention in human diseases.

# 2. Proposed physiological functions of CDK5

## 2.1. CDK5 in the immune response

CDK5 has been postulated to regulate different aspects of the immune response ranging from survival and motility of lymphocytes to cytokine secretion by lymphocytes, neutrophils and macrophages (figure 1b). Using mice reconstituted with hematopoietic progenitors from Cdk5$^{-/-}$ embryos, Pareek et al. [35] demonstrated that during T-cell activation, Cdk5 phosphorylates actin modulator coronin 1a on Thr418 and regulates actin polarization, thereby promoting T-cell survival and motility. In addition, CDK5 has also been shown to augment the production of interleukin 2 (IL-2) by T cells, by impeding the repression of IL-2 transcription by the Histone deacetylase 1 (HDAC1) repressor complex [36]. Specifically, CDK5 phosphorylates mSin3a protein, an essential component of HDAC1 complex at Ser861 and prevents

binding of the complex to IL-2 promoter, thereby upregulating IL-2 production [36] (figure 1b). Consistent with this model, Cdk5$^{-/-}$ T cells were shown to be deficient in IL-2 production, leading to increased susceptibility of mice to infections [36]. Furthermore, transplantation of Cdk5-null bone marrow stem cells and T cells, derived from C57BL/6 J mice into lethally irradiated B6D2F1 mice, resulted in a significant reduction in graft-versus-host disease (GVHD) [36]. The authors attributed these manifestations to decreased T-cell migration to secondary lymphoid organs, reduced proliferation of T cells within these organs and the diminished numbers of cytokine-producing donor T cells [37]. Collectively, these data suggest that inhibiting CDK5 function may mitigate GVHD in leukaemia patients who have undergone allogeneic bone marrow or stem cells transplant.

In addition, CDK5 was shown to regulate secretion of cytokines by neutrophils in response to injury or infection [37]. This function has been attributed to the ability of CDK5 to phosphorylate Ser56 residue of vimentin. The phosphorylation of vimentin at Ser56 results in vimentin depolymerization and disassembly of neutrophil cytoskeleton, thereby allowing the release of neutrophil vesicular contents. Conversely, inhibition of CDK5 activity with pan-CDK inhibitors olomoucine or roscovitine leads to a decrease in the phosphorylation of Ser56 of vimentin and reduced neutrophil secretion [38]. Lastly, in macrophages, CDK5 was proposed to inhibit the production of interleukin 10 (IL-10), an anti-inflammatory cytokine secreted in response to stimulation by lipopolysaccharides [39].

## 2.2. CDK5 in regulation of insulin levels

Several studies have shown that CDK5 serves to decrease insulin production in response to chronically high glucose levels. Knockdown of CDK5 or inhibition of CDK5 activity with CDK5 inhibitory peptide or with pan-CDK inhibitors roscovitine or olomoucine in pancreatic β cell lines enhanced insulin secretion in high-glucose conditions [15,40,41]. Consistent with this observation, pancreatic β cells isolated from p35 knockout mouse also showed increased insulin secretion in response to high glucose levels. Interestingly, no effect was seen when the glucose level was low.

However, when roscovitine or olomoucine were combined with pharmacological compounds that increase endogenous insulin secretion such as glibenclamide, sulfonylurea or glucagon-like peptide 1 (GLP-1), CDK5 inhibition had an additive effect on insulin secretion in both high-glucose and low-glucose conditions [42,43].

Mechanistically, Ubeda et al. have shown that in high-glucose conditions, CDK5 inhibits the binding of a transcription factor pancreatic duodenal homeobox 1 (PDX-1) to insulin promoter and promotes nuclear export of PDX-1. The inhibition of CDK5 activity in high-glucose conditions maintains the binding of PDX-1 to the insulin gene promoter, thereby increasing insulin production [41]. Another study reported an increase in nuclear PDX-1 levels upon inhibition of CDK5 with myricetin, a natural flavonoid that binds to the ATP binding pocket of CDK5. According to these authors, this protects pancreatic β cells from high-glucose-induced apoptosis [44].

In contrast with these studies, Daval & Gurlo [45] demonstrated that depletion of Cdk5 with siRNA or inhibition of Cdk5 kinase using roscovitine increased the apoptotic rate

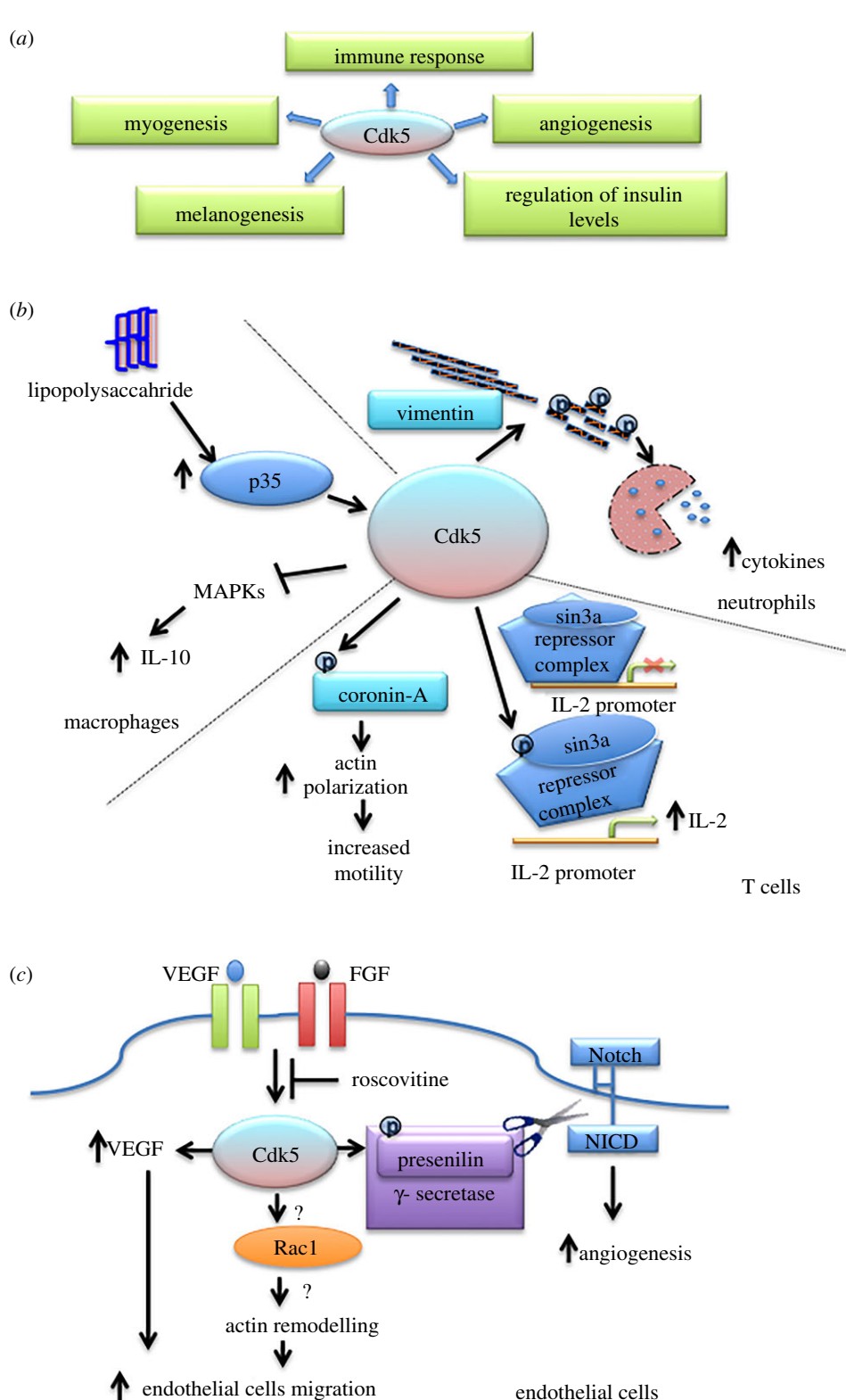

**Figure 1.** Proposed physiological functions of Cdk5. (*a*) A summary of non-neuronal functions of Cdk5. (*b*) Cdk5 phosphorylates actin modulator coronin 1a and regulates actin polymerization to promote T-cell survival and motility. In T cells, Cdk5-driven phosphorylation of Sin3a, a component of HDAC1 repressor complex, promotes IL-2 production by impeding the transcriptional repression of the *IL-2* gene. In addition, Cdk5 phosphorylates vimentin leading to cytoskeletal rearrangements, thereby facilitating the secretion of cytokines by neutrophils. In response to stimulation by lipopolysaccharides, Cdk5 inhibits the activation of mitogen-activated protein kinases (MAPKs) and subsequent secretion of IL-10 by macrophages. (*c*) Cdk5 and angiogenesis: Cdk5 is activated by proangiogenic factors, such as VEGF and FGF, and regulates angiogenesis by affecting Rac-mediated actin remodelling and endothelial migration. In addition, Cdk5 can directly phosphorylate VEGF and regulate VEGF levels; inhibition of Cdk5 activity decreases VEGF levels. Furthermore, the phosphorylation of presenilin, a γ-secretase subunit, by Cdk5 increases the cleavage of Notch1 to NICD and promotes Notch signalling and angiogenesis.

of rat insulinoma INS 822/13 cells by inhibiting FAK/AKT survival pathway. Moreover, decreased phosphorylation of a well-established CDK5 substrate FAK at Ser727 has been

observed in β cells of diabetic rats and in humans with type 2 diabetes. The authors postulated that Cdk5 phosphorylates FAK at Ser727, which in turn activates PI3 K/AKT pathway

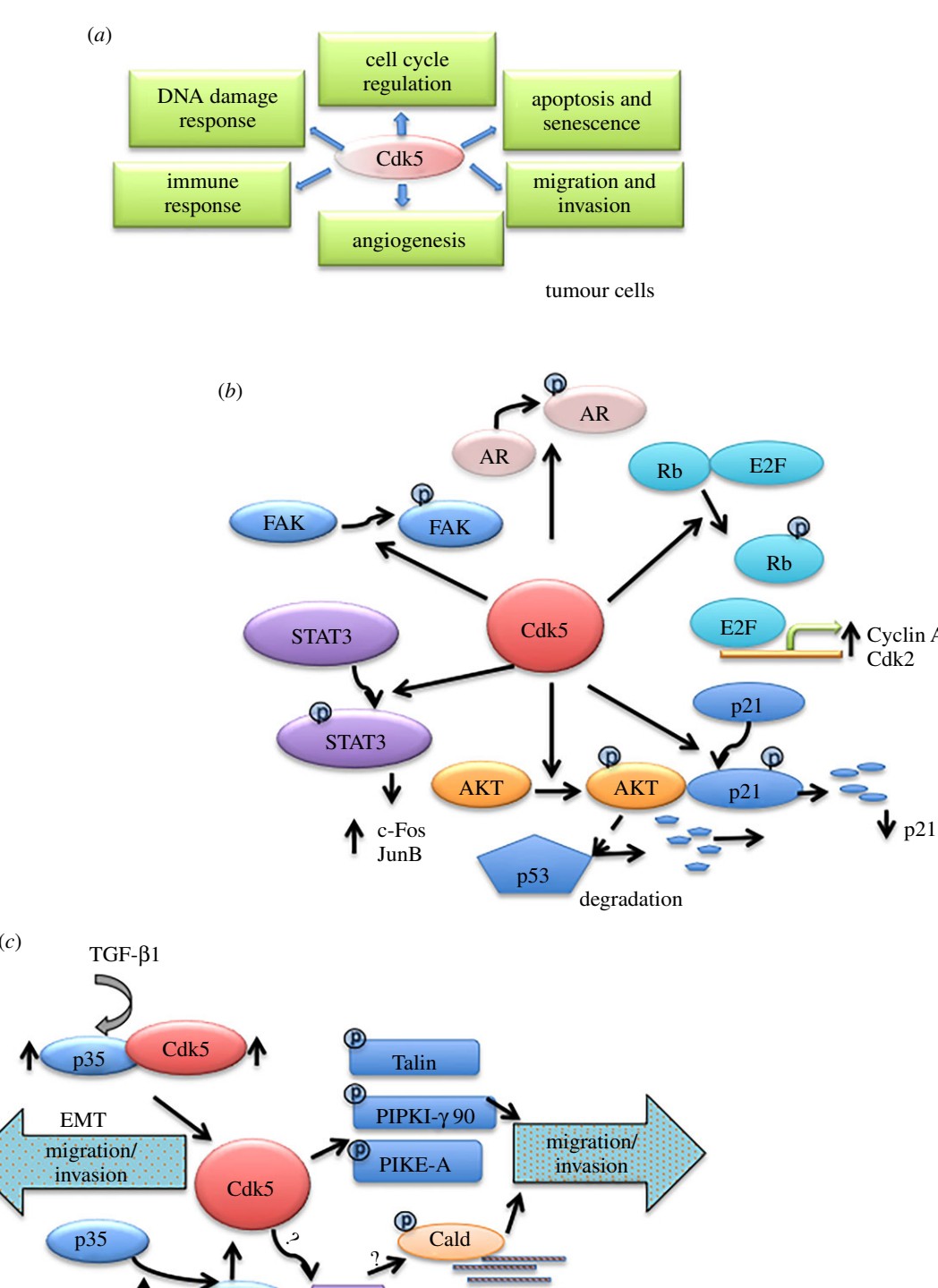

**Figure 2.** Proposed functions of CDK5 in cancer cells. (*a*) A summary of different CDK5 functions in cancer cells. (*b*) CDK5 and cell cycle regulation: CDK5 has been shown to phosphorylate RB1, STAT3, AR, FAK, AKT, p21^Cip1 and each of these events has been linked to stimulation of cell cycle progression and proliferation by CDK5. (*c*) CDK5 and migration and invasion: in pancreatic ductal adenocarcinoma, K-Ras promotes the cleavage of p35 to p25, resulting in CDK5 hyper-activation. This, in turn, promotes migration and invasion through RalA-GTP. Likewise, in breast cancer cells, TGF-β1 promotes the cleavage of p35 to p25 and activates CDK5, resulting in EMT. In addition, CDK5 may promote migration of cancer cells by directly phosphorylating talin, FAK, PIPKIγ90, or GTPase PIKE-A, or actin regulatory protein caldesmon.

and protects β cells from apoptosis [45]. In addition, Choi *et al.* demonstrated that Cdk5 is activated in high-fat diet-induced obese mice. This leads to insulin resistance, a condition characterized by impaired response to insulin that results in elevated levels of blood glucose [46]. The authors showed that in these obese mice, elevated levels of local and circulating cytokines such as IL-6 or TNF-α promote the cleavage of p35 to p25, resulting in Cdk5 hyper-activation. This, in turn, leads to the phosphorylation of

peroxisome proliferator-activated receptor γ (PPARγ) at serine 273. PPARγ, a nuclear receptor, is necessary for differentiation of fibroblastic precursors into adipocytes and regulates the expression of several metabolic regulators and adipokines including insulin-sensitizing hormone, adiponectin. The phosphorylation of PPARγ at Ser273 by Cdk5 decreases the expression of adiponectin through an unknown mechanism, leading to development of insulin resistance. Potent synthetic PPARγ ligands such as MRL24 or

rosiglitazone (both widely used as insulin-sensitizing anti-diabetic drugs) were proposed to shield Ser273 residue from phosphorylation by Cdk5, thereby restoring expression of adiponectin and overcoming insulin resistance [46]. Interestingly, the same group later showed that when Cdk5 expression was ablated in the adipose tissue of obese mice, the phosphorylation of Ser273 PPARγ residue was actually increased when compared with Cdk5 wild-type mice. Consequently, ablation of Cdk5 did not alleviate insulin resistance. The authors proposed that in the absence of Cdk5, a compensatory phosphorylation of Ser273 PPARγ is carried out by the ERK kinase, and this event has the same functional consequences as the phosphorylation of Ser273 by Cdk5. Hence, pharmacologic inhibition of Cdk5 may be ineffective in the obese mice, and MEK and ERK inhibition may provide a better strategy to combat diabetes [47].

## 2.3. CDK5 in angiogenesis

Several studies proposed different roles for CDK5 in angiogenesis (figure 1c). According to Liebl et al. [48], CDK5 promotes endothelial cell migration and angiogenesis by remodelling the actin cytoskeleton via GTPase Rac1. This conclusion was based on the observation that inhibition of CDK5 by roscovitine impeded migration of endothelial cells and completely blocked induction of vessel formation by angiogenic growth factors such as fibroblast growth factor (FGF) or vascular endothelial growth factor (VEGF) in vitro and in vivo. In addition, mouse cornea neovascularization upon FGF treatment was blocked following intraperitoneal administration of roscovitine. This effect was attributed to reduced lamellipodia formation in the leading edge of migrating cells due to disruption in the localization of Rac1 and its effector cortactin [48]. Another group suggested that CDK5 might affect angiogenesis by regulating VEGF levels. The authors showed that depletion of Cdk5 using siRNA, or inhibition of Cdk5 activity with roscovitine, decreased the expression of VEGF in rat pituitary cell lines [49].

Merk et al. [50] proposed that CDK5 phosphorylates and activates presenilin, a catalytic subunit of γ-secretase. This stimulates proteolytic cleavage of Notch1 to Notch1 intracellular domain (NICD) by γ-secretase, and activates Notch-dependent signalling, which represents a key angiogenesis-promoting pathway. Indeed, depletion of CDK5 or inhibition with roscovitine decreased phosphorylation and total levels of presenilin, and diminished NICD levels. This was accompanied by a decrease in the expression of Notch downstream target genes [50].

Paradoxically, endothelial-specific Cdk5 ablation in mice led to hypervascularization of the yolk sacs, resulting in perinatal lethality. The authors found that Cdk5 ablation promoted non-productive angiogenesis characterized by an increase in vascularity and a decrease in tissue perfusion, a phenotype similar to that observed in mice with defective Notch signalling [50]. The same group showed, also using conditional Cdk5 knockout mice, that Cdk5 is required for development of lymphatic vessels and formation of lymphatic valves. Endothelial-specific ablation of Cdk5 resulted in defective lymphatic vessel patterning and valve formation in embryos, leading to lymphatic dysfunction and lymphedema. The authors proposed that Cdk5 phosphorylates Foxc2, a transcription factor required for valve formation

and lymph vessel development. This, in turn, promotes the recruitment of Foxc2 to chromatin and stimulates expression of Foxc2 target genes such as Epb41l5 and Csnk1g3 [51].

## 2.4. CDK5 in myogenesis

Early indication that CDK5 is involved in myogenesis came from the study of Xenopus embryos. These studies demonstrated that Cdk5, together with the Xenopus orthologue of p35, Xp35.1, regulates the expression of transcription factors involved in myogenesis, such as MyoD and myogenic regulator factor 4 (MRF4). Inhibition of Cdk5 activity by injection of a dominant-negative Cdk5 construct into early Xenopus embryos resulted in decreased expression of MyoD and MRF4 and disruption of somitic muscles [52]. Since then, several studies have further documented the requirement for CDK5 activity during myogenic differentiation. Using primary myoblasts and immortalized myoblast cells, De Thonel et al. demonstrated that at an early stage of muscle development, protein kinase Cζ (PKCζ) phosphorylates p35 on Ser-33. This allows calpain-mediated cleavage of p35 to a more stable p25 form, resulting in the activation of CDK5 [53]. Active p25–CDK5 complexes phosphorylate nestin, the class VI cytoskeletal intermediate filament protein, at Thr316, leading to reorganization of nestin filaments during myogenic differentiation [54,55]. Conversely, inhibition of CDK5 activity either using roscovitine or by expression of dominant-negative CDK5 blocked the phosphorylation of nestin at Thr316, thereby impairing reorganization of nestin filaments and inhibiting myoblast differentiation [55].

## 2.5. CDK5 in melanogenesis and epidermal development

Recently, CDK5 was shown to be essential for normal epidermal development and melanogenesis. Dong et al. demonstrated that in alpaca melanocytes, depletion of Cdk5 decreased melanin production by downregulating the expression of essential components of the melanogenesis pathway, such as transcription factor paired box 3 (PAX3), melanocortin receptor type I (MC1R) and tyrosinase (TYR) [56]. The authors observed that in vivo depletion of Cdk5 resulted in lighter coat colour and polarized distribution of melanin. In addition, they observed an abnormally thickened epidermis and ascribed this phenotype to decreased levels of keratin 10 upon Cdk5 depletion [56]. While this study provided a strong evidence of involvement of CDK5 in normal epidermal development and melanogenesis, further work is needed to fully understand how CDK5 affects these processes by regulating transcription factors and enzymes involved in melanogenesis.

## 2.6. Other proposed physiological functions of CDK5

CDK5 was shown to inhibit the motility of corneal epithelial cells by regulating stress fibre formation and contraction in migrating cells [57]. According to Tripathi & Zelenka [57], p39–CDK5 complex stabilizes stress fibres and inhibits migration of epithelial cells by suppressing the activity of Src, thereby preventing Src-mediated phosphorylation and activation of RhoGAP, an upstream inhibitor of Rho. This,

in turn, augments the Rho-ROCK signalling-dependent phosphorylation of myosin, an essential event for stress fibre formation and contraction of cells [57]. It was also shown that CDK5 directly phosphorylates a scaffold protein muskelin that binds to myosin and facilitates the phosphorylation of myosin by the Rho-ROCK signalling. This event further stabilizes stress fibres [58]. Consistent with this model, inhibition of CDK5 activity by roscovitine or oloumoucine decreased the phosphorylation of myosin regulatory light chain (MRLC), a component of the myosin complex, resulting in disrupted stress fibre organization and increased migration in corneal epithelial cells [57,58].

# 3. Proposed roles of CDK5 in cancer

Increased expression of CDK5, p35 or p39 and the resulting hyper-activation of CDK5 have been reported in pancreatic, medullary thyroid, non-small cell lung, small cell lung, colorectal, liver, breast and ovarian cancers, glioblastoma multiforme, multiple myeloma and mantle cell lymphoma [40,59–66]. In case of pancreatic ductal adenocarcinomas and non-small lung cell cancers, increased CDK5 expression was attributed to the amplification of the CDK5 gene [67,68]. High expression of CDK5 correlates with poor prognosis and shorter patient survival in non-small cell lung, small cell lung, ovarian, colorectal and breast cancers, as well as in multiple myeloma [61,62,69–71]. Collectively, these observations indicate that CDK5 may act as an oncogene. However, other authors reached an opposite conclusion. Thus, Sun et al. [72] and Cao et al. [73] proposed that CDK5 represents a tumour suppressor in gastric cancer. The authors reported that low levels of CDK5 in gastric carcinomas correlated with poor patient survival and the presence of metastases, while high CDK5 levels conferred an improved prognosis. Similarly, Lu et al. [74] reported that in hepatocellular carcinoma, patients with lower expression of p39 in the tumours displayed poorer overall survival, when compared with patients with high p39 expression. Another indication of an involvement of CDK5 is cancer pathogenesis is provided by the observations linking single-nucleotide polymorphism (SNP) in the promoter region of CDK5 gene to increased susceptibility to lung cancer in the Korean population [75] and higher risk of prostate cancer among African-Americans [76].

CDK5 has been implicated to play a role in various aspects of tumorigenesis and tumour progression (figure 2a). In the following section, we will discuss these different findings.

## 3.1. Role of CDK5 in cell cycle

The function of CDK5 in cancer cell proliferation remains controversial. Pozo et al. [77] reported that transgenic mice engineered to overexpress p25 (a hyperactive form of p35) in thyroid C cells developed medullary thyroid carcinoma (MTC). Importantly, tumour growth was dependent on p25, as switching off p25 expression halted MTC progression [77]. Consistent with the notion that CDK5 promotes cell proliferation, inhibition of CDK5 activity using roscovitine, olomoucine, dinacilcib, CP681301 or expression of dominant-negative CDK5 constructs, or depletion using anti-CDK5 siRNAs decreased proliferation of in vitro cultured MTC, prostate, colon, colorectal and liver cancer cells, as

well as proliferation of tumour cells in xenografts in vivo [59,62,77–79]. However, some authors have argued that CDK5 inhibits tumour cell proliferation and hence it displays growth-suppressive properties [73,80,81], while others concluded that CDK5 does not play a role in regulating proliferation [59,82,83], Below, we provide a summary of the proposed molecular mechanisms through which CDK5 may regulate cell proliferation (figure 2b).

CDK5 was postulated to phosphorylate the retinoblastoma protein (RB1). The classical consequence of RB1 phosphorylation by cyclin–CDK complexes is the release of E2Fs transcription factors and the resulting transcription of genes required for cell cycle progression [84]. Pozo et al. demonstrated that MTC tumours arising in p25 transgenic mice displayed increased phosphorylation of RB1 at Ser807/811 as well as increased expression of an E2F target, cyclin A as well as Cdk2. Turning off p25 expression in these mice decreased the phosphorylation of RB1 at Ser 807/811 and decreased expression of Cdk2 and cyclin A [77]. A similar effect was seen upon treatment of MTC cell lines with a Cdk5 inhibitor CP681301 [77]. Analogous conclusions were independently reached by Meder et al. [22,85].

Other groups also reported that inhibition of CDK5 activity in MTC and prostate cancer cell lines decreased cell proliferation. This effect was attributed to the ability of CDK5 to phosphorylate STAT3 at Ser727, and upregulate the expression of STAT3 target genes, c-Fos and JunB. The authors reported that chemical inhibition of CDK5 activity decreased STAT3 Ser727 phosphorylation and diminished expression of c-Fos and JunB [78,86,87].

Rea et al. proposed that CDK5 is required for proper mitotic spindle assembly and chromosome alignment during mitosis. This function was attributed to phosphorylation of FAK by CDK5 at Ser732. Phosphorylated FAK localizes to the microtubules of the spindle and facilitates microtubule depolymerization. The inhibition of CDK5 activity with roscovitine decreased the phosphorylation of FAK at Ser732 and impaired mitotic spindle assembly and correct chromosome alignment during mitosis [88].

Lindqvist et al. proposed that CDK5 promotes proliferation of prostate cancer cells by regulating p53 and p21$^{Cip1}$ levels via an indirect mechanism. Specifically, they reported that CDK5 phosphorylates AKT at Ser 473, thereby activating its kinase activity [89]. Active AKT phosphorylates and activates MDM2, a ubiquitin ligase which targets p53 for ubiquitin-mediated degradation [90]. This, in turn, results in decreased levels of a CDK inhibitor p21$^{Cip1}$. Consistent with these observations, several groups reported that siRNA-mediated depletion of CDK5 led to increased p53 and p21$^{Cip1}$ levels, resulting arrest of cells in G1 phase of the cell cycle. Huang et al. demonstrated that CDK5 can directly phosphorylate p21$^{Cip1}$ and regulate p21$^{Cip1}$ protein levels. The authors reported that phosphorylation of p21$^{Cip1}$ Ser130 residue by CDK5 promotes proteasomal degradation of p21$^{Cip1}$. This, in turn, activates CDK2 and accelerates cell cycle progression of cancer cells [91].

In contrast with all these findings, CDK5 was postulated by others to act as a tumour suppressor in gastric cancers. Cao et al. [73] reported decreased levels of CDK5 in gastric tumours when compared with normal adjacent tissue. Importantly, ectopic expression of CDK5 in gastric cancer cells decreased tumour cell proliferation in vitro and in vivo in xenografts. The authors proposed that nuclear CDK5

royalsocietypublishing.org/journal/rsob    Open Biol. **10**: 190287

royalsocietypublishing.org/journal/rsob    Open Biol. **10**: 190287

binds to E2F1 and attenuates the E2F1–DNA interaction, thereby inhibiting cell proliferation. In addition, nuclear CDK5 can upregulate the levels of a CDK inhibitor p16$^{INK4a}$, which further contributes to cell cycle arrest [73].

## 3.2. CDK5 in apoptosis

The role of CDK5 in cancer cell survival remains unclear, with some groups reporting that CDK5 prevents while others proposing that CDK5 promotes apoptosis. Different molecular mechanisms were postulated to mediate the function of CDK5 in tumour cell survival. CDK5 was implicated to protect breast cancer cells against reactive oxygen species (ROS)-mediated apoptosis. The depletion of CDK5 using siRNA leads to prolonged opening of mitochondrial permeability transition pores (mPTP) by an unknown mechanism, which results in mitochondrial depolarization followed by an increase in ROS levels and ultimately caspase activation and apoptosis [92]. Other authors postulated that in hematopoietic malignancies, in glucose-rich conditions, CDK5 phosphorylates BH3-only protein Noxa at Ser12 leading to its sequestration in the cytosol, which in turn inhibits the mitochondrial pro-apoptotic pathway. According to this model, glucose deprivation or inhibition of CDK5 results in dephosphorylation of Noxa and translocation of Noxa to mitochondria leading to tumour cell apoptosis [93].

In addition, CDK5 was postulated to regulate the survival of breast cancer tumour initiating cells possibly by phosphorylating Forkhead box Type O transcription factor 1 (Foxo1), thereby promoting its nuclear export. This, in turn, suppresses the expression of pro-apoptotic genes such as Bim, which represent Foxo1 transcriptional targets. According to this model, depletion of CDK5 increases the levels of nuclear Foxo1, which leads to increased expression of Bim protein and tumour cell apoptosis [94]. Another group demonstrated that in podocytes and neurons, CDK5 can physically interact with a cyclin-like protein called cyclin I. Cyclin I–CDK5 complexes prevent injury-related apoptosis of podocytes and neurons by increasing the expression of pro-survival proteins such as BCL2 or BCL-XL [95]. Interestingly, CDK5 was also shown to be activated by binding to cyclin I in cisplatin-resistant cervical cancer cells, where cyclin I–CDK5 complex was proposed to protect tumour cells from apoptosis. The expression levels of CDK5 and cyclin I are higher in cisplatin-resistant cancers compared to cisplatin-sensitive ones and correlate with worse survival. Importantly, depletion of CDK5 using siRNA sensitized cervical cancer cells to cisplatin [96]. Hence, inhibition of CDK5 in combination with cisplatin may represent an effective therapeutic strategy.

In contrast to the above-described findings, Lee *et al.* [97] reported that CDK5 promotes apoptosis in neuroblastoma cell lines. The authors demonstrated that upon mitomycin C-induced DNA damage or sodium nitroprusside-induced oxidative stress, CDK5 phosphorylates p53 at multiple residues (Ser15, Ser33, Ser46). The phosphorylation of p53 by CDK5 disrupted the interaction of p53 with E3 ubiquitin ligase MDM2, thereby preventing ubiquitin-mediated degradation of p53. This, in turn, stabilizes nuclear p53 resulting in transactivation of pro-apoptotic genes [97]. Consistent with this model, inhibition of CDK5 activity using siRNAs or dominant-negative constructs inhibited mitomycin C- or sodium nitroprusside-induced apoptosis [97]. Similar

observations were made in lung, cervical and prostate cancer cell lines [73,80,81].

## 3.3. CDK5 in senescence

Ectopic expression of the retinoblastoma protein (RB1) in RB1-null cancer cell lines results in senescence-like phenotype [98]. It was demonstrated that expression of RB1 increases p35 levels, thereby upregulating the activity of CDK5 [99]. Active CDK5 promotes senescence either by repressing Rac1 activity or by upregulating the expression of F-actin or of an F-actin associated protein, Ezrin [99,100]. Regardless of the exact molecular mechanism, the increase in CDK5 activity leads to rearrangement of actin filaments resulting in a characteristic flattened appearance of senescent cells [99,100]. Importantly, the senescence phenotype and the associated molecular events (repression of Rac1, increased levels of F-actin and Ezrin) were reversed by knocking out the p35 (*CDK5R1*) or *CDK5* genes or by inhibition of CDK5 activity with roscovitine [99,100].

## 3.4. CDK5 and androgen receptor function

Hsu *et al.* [101] have shown a positive correlation between protein levels of the androgen receptor (AR) and those of CDK5 or p35 in prostate cancer specimens. This group demonstrated that in prostate cancer cells, CDK5 directly phosphorylates the AR at Ser81. In addition, CDK5 phosphorylates STAT3 at Ser 727. Both phosphorylation events promote the binding of STAT3 to AR, resulting in AR stabilization, which stimulates proliferation of prostate cancer cells [101]. Other groups reported that CDK5 phosphorylates the AR on another site (Ser308). This stabilizes AR independently of STAT3. Regardless of the exact mechanism, stabilization of AR enhances transcription of AR-induced genes that promote prostate carcinogenesis [86,89].

## 3.5. Migration and invasion

Several studies have implicated CDK5 in regulating migration and invasiveness of cancer cells [60,65,102] (figure 2*c*), but the outcome (stimulation versus inhibition of migration) as well as the molecular mechanism remains unclear. Depletion of CDK5 or inhibition of its activity in several *in vitro* cultured cancer cell lines resulted in reorganization of microtubules and the loss of cell polarity [82]. This was accompanied by destabilization of focal adhesions and suppression of podosomes and invadopodia formation, which ultimately led to decreased migration, invasion and anchorage-independent growth [103]. Moreover, depletion of CDK5 in xenografts reduced lungs and livers [103]. Eggers *et al.* reported that CDK5 promotes migration and invasion specifically in K-Ras mutant pancreatic ductal adenocarcinoma. The authors demonstrated that mutant K-Ras stimulates the cleavage of p35 to a more stable p25 species. This leads to hyper-activation of CDK5 kinase activity, resulting in increased tumour cell migration and invasion. Inhibition of CDK5 by ectopic expression of dominant-negative CDK5 constructs, or treatment of *in vitro* cultured cells with roscovitine significantly decreased the migration and invasion of pancreatic cancer cells [67]. Feldmann *et al.* [59] reported that inhibition of CDK5 led to decreased activation of crucial effectors of Ras signalling pathway, RalA-GTP and RalB GTP. Based on these observations, they postulated that in

pancreatic cancer cell lines, CDK5 operates downstream of Ras. However, the precise mechanism of how CDK5 regulates the activity of RalA/RalB remains unclear. In addition to these studies, other authors postulated that CDK5 may promote migration of cancer cells by phosphorylating an actin regulatory protein caldesmon, or talin, or focal adhesion kinase FAK, PIPKIγ90 or GTPase PIKE-A [60,103–107]. Moreover, depletion of CDK5 or inhibition of CDK5 kinase activity was shown to reduce the expression of mesenchymal markers α-smooth muscle actin (α-SMA), N-cadherin, β-catenin and vimentin and to increase the expression of an epithelial marker E-cadherin [63,104], thereby impeding cell migration and invasion. Conversely, overexpression of CDK5 was reported to upregulate N-cadherin, α-SMA and β-catenin levels [66,108,109]. Liang *et al.* [63] postulated that CDK5 induces epithelial to mesenchymal transition and promotes migration and invasion by phosphorylating focal adhesion kinase (FAK). The authors observed that stimulation of *in vitro* cultured breast cancer cells with transforming growth factor β1 (TGF-β1) upregulated the expression of CDK5 and of p35 leading to an increased CDK5 activity. Furthermore, they proposed that activated CDK5 phosphorylates FAK at Ser 732 and Tyr397 residues, and this phosphorylation is important for increased expression of α-SMA and for migration and invasion of breast cancer cells [63].

In contrast with these findings, several groups reached an opposite conclusion, namely that CDK5 inhibits cancer cell motility. This inhibitory function has been ascribed to the ability of CDK5 to phosphorylate a tumour suppressor protein DLC1 [71], a transmembrane glycoprotein podoplanin (PDPN) [110] or histone methyltransferase EZH2 [111]. Jin *et al.* demonstrated that CDK5 inhibits pancreatic cell migration and invasion by phosphorylating EZH2 at Thr261. This promotes binding of phosphorylated EZH2 to the F-box and WD repeat domain-containing 7 (FBW7) protein, a substrate recognition component of Skp1–Cul1–F-box (SCF) ubiquitin complex, resulting in ubiquitin-mediated degradation of EZH2 [111]. EZH2, a component of the polycomb repressive complex 2 (PRC2), was previously shown to promote pancreatic cancer cell migration and invasion by suppressing the expression of E-cadherin [112]. Conversely, inhibition of CDK5 function (by expression of dominant-negative CDK5) resulted in increased levels of EZH2, decreased levels E-cadherin level and stimulated cell migration and invasion [111,112]. Hence, the role of CDK5 in promoting or inhibiting cancer cell migration remains controversial and requires further studies.

## 3.6. CDK5 in angiogenesis

The formation of new blood vessels provides tumour cells with necessary nutrients and oxygen and facilitates tumour growth. Several studies postulated that CDK5 promotes tumour angiogenesis [48,50,51]. Analyses of human hepatocellular carcinoma (HCC) tissue array revealed that overexpression of CDK5 correlates with higher vascular density. At the molecular level, CDK5 was shown to phosphorylate an angiogenic factor hypoxia-inducible factor (HIF-1α) at Ser687, thereby protecting it from proteasomal degradation. Inhibition of Cdk5 activity by injection of roscovitine to mice bearing tumours derived from a hepatocellular carcinoma cell line resulted in decreased tumour vascularization. In addition, depletion or inhibition of CDK5 with

roscovitine in hepatocellular carcinoma cell lines decreased expression of HIF-1α levels, and decreased expression levels of HIF-1α targets VEGFA, EphrinA1 *in vivo* and *in vitro* [113]. Consistent with these findings, inhibition of CDK5 reduced tumour growth and improved sensitivity to anti-angiogenic treatment in U87 glioblastoma and Lewis lung carcinoma xenografts [50]. The ability of CDK5 to promote angiogenesis provides an exciting prospect of targeting CDK5 in anti-angiogenic therapy, possibly in combination with the currently used anti-angiogenic compounds.

## 3.7. CDK5 in DNA damage response

There is increasing evidence that CDK5 may play a role in the initiation of DNA damage response (DDR) in cancer cells. Ehrlich *et al.* demonstrated an increased levels of CDK5 and upregulation of CDK5 kinase activity upon exposure of cancer cells to DNA damage-inducing agents such as ionizing radiation, topoisomerase inhibitors or PARP inhibitors. CDK5 was shown to phosphorylate ataxia telangiectasia mutated (ATM) protein at Ser729, thereby activating the DDR pathway in hepatocellular carcinoma cell lines [66,114]. Others proposed that CDK5 promotes DNA repair by phosphorylating and activating the enzymatic activity of Poly (ADP-ribose) Polymerase 1 (PARP), an enzyme involved in single-strand break repair [115,116] or by phosphorylating and activating STAT3, leading to upregulation of Eme1, an endonuclease that is implicated in processing of broken replication forks [117,118], or that it phosphorylates replication protein A 32 (RPA32) on Serine 23, 29 and 33, which represents a required priming event for intra-S phase checkpoint activation and DNA repair [119]. All these mechanisms are expected to augment DNA damage repair. Indeed, inhibition of CDK5 kinase activity or depletion of CDK5 using siRNAs increased sensitivity of cancer cells to DNA damaging agents [114,117,118,120].

## 3.8. CDK5 in anti-tumour immune response

CDK5 may play a role in the anti-tumour immune response, by regulating the levels of programmed cell death ligand 1 (PD-L1) in tumour cells. Dorand *et al.* [83] demonstrated that treatment of medulloblastoma cell lines with interferon γ (IFNγ) causes upregulation of p35 levels, resulting in increased CDK5 kinase activity. The authors proposed that CDK5 directly or indirectly inhibits kinases responsible for phosphorylating IRF2BP2, a co-repressor of interferon regulatory factor (IRF2) complex, leading to de-repression of PD-L1 expression. This, in turn, helps medulloblastomas to evade immune elimination. Consistent with this model, depletion of CDK5 using shRNA led to hyperphosphorylation of IRF2BP2, increased expression of IRF2 and decreased PD-L1 levels. The depletion of CDK5 in medulloblastoma xenograft model resulted in diminished PD-L1 expression and increased the numbers of CD4[+] tumour-infiltrating lymphocytes, resulting in robust CD4[+] T-cell-mediated tumour rejection [83].

## 3.9. 'CDK5-specific' inhibitors

The emerging evidence implicating CDK5 in several pathological processes, such as neurodegeneration, diabetes and cancer, points to CDK5 as an attractive therapeutic target.

royalsocietypublishing.org/journal/rsob Open Biol. 10: 190287

**Table 1.** Compounds used to inhibit CDK5.

| compound | targets | IC$_{50}$ (µM) | references |
|---|---|---|---|
| R-roscovitine | CDK1 | 0.65 | [120–123] |
| | CDK2 | 0.7 | |
| | CDK4 | >100 | |
| | CDK5 | 0.16 | |
| | CDK6 | >100 | |
| | CDK7 | 0.46 | |
| | CDK8 | >100 | |
| | CDK9 | 0.6 | |
| dinaciclib | CDK1 | 0.003 | [124] |
| | CDK2 | 0.001 | |
| | CDK5 | 0.001 | |
| | CDK9 | 0.004 | |
| flavopiridol | CDK1 | 0.003 | [124] |
| | CDK2 | 0.012 | |
| | CDK5 | 0.014 | |
| | CDK9 | 0.004 | |
| olomoucine | CDK1 | 7 | [125] |
| | CDK2 | 7 | |
| | CDK5 | 3 | |
| | CDK6 | 180 | |
| purvalanol A | CDK1 | 0.075 | [126] |
| | CDK5 | 0.004 | |
| CP681301 | CDK5 | 0.5 | [127] |
| CP668863 | CDK2 | 0.006 | [128] |
| | CDK5 | 0.008 | |
| PJB | CDK2 | 0.098 | [103] |
| | CDK5 | 0.064 | |

Unfortunately, no CDK5-specific inhibitors are currently available. Therefore, most of the studies conducted to date employed non-specific pan-CDK inhibitors to inhibit CDK5, However, as we describe below, these compounds inhibit several other kinases, in addition to CDK5 (table 1).

Flavopiridol, often used to inhibit CDK5, represents an ATP-competitive inhibitor of CDK1, CDK2, CDK5 and CDK9 [124] (table 1). Because of its ability to inhibit cell cycle CDKs, it has been used in Phase II clinical trials for the treatment of several solid tumours types. The studies reported a narrow therapeutic window and low efficacy. The other widely used 'CDK5 inhibitors' (R)-roscovitine and olomoucine inhibit CDK2, CDK1, CDK7 and CDK9, in addition to CDK5 [120–123,125] (table 1). These compounds were used in phase I clinical studies, again due to their ability to inhibit cell cycle CDKs, for the treatment of advanced solid tumours but did not show any significant response. Currently, there is one ongoing clinical trial with roscovitine in Cushing disease and another trial in combination with a chemotherapeutic compound sapacitabine, for the treatment of advanced solid tumours [129,130]. Another 'CDK5 inhibitor', dinaciclib, inhibits CDK1, CDK2, CDK5 and CDK9 [124]

(table 1). Dinaciclib was reported to have a 10-fold higher therapeutic index than flavopiridol [124]. This compound is currently being used in three clinical trials for patients with different tumour types, involving combination therapies with venetoclax, veliparib and pembrolizumab [131–133]. Other small molecule inhibitors such as CP668863, CP681301 and N-(5-isopropylthiazol-2-yl)-3-phenylpropanamide (PJB) show increased selectivity for CDK5 and CDK2 over other CDKs [103,127,128], but the efficacy of these inhibitors in vivo remains to be tested.

Given the non-specific properties of all these inhibitors, there is an urgent need to develop compounds that potently and selectively inhibit CDK5. Such compounds may be efficacious in the treatment of various diseases, such as neurodegeneration, diabetes and cancer.

It should be noted that an alternative strategy has been developed to interfere with CDK5 activity. Thus, it was found that a peptide containing 125 amino acid residues of p35, called CDK5 inhibitory peptide (CIP), specifically inhibits hyper-activation of CDK5 by p25, while having no effect on physiological activation of CDK5 by p35/p39 [15]. An even smaller 24 amino acid derivative of p25 was found to display a greater inhibitory potential than CIP [134]. These inhibitors reduced hyper-activation of Cdk5 by p25 in vivo and prevented the onset of AD and ALS in mouse models [134]. In addition, Zheng et al. [15] reported that in high-glucose conditions, CIP inhibits p25-mediated Cdk5 hyper-activation leading to the recovery of insulin secretion. These studies suggest that CIP may serve as a novel therapeutic agent for the treatment of several neurodegenerative diseases as well as type 2 diabetes.

## 4. Conclusion

Over the last two decades, several non-neuronal functions of CDK5 have been proposed. It seems likely that in addition to development and normal physiology of the nervous system and pathogenesis of neurodegenerative diseases, CDK5 plays distinct roles in physiology and pathogenesis of several different tissues. However, most of the non-neuronal functions of CDK5 remain controversial. For instance, while several studies proposed that CDK5 acts as an oncogene and it promotes proliferation, migration and invasion of cancer cells, others reported that CDK5 functions as a tumour suppressor by promoting apoptosis or inhibiting proliferation, and that it impedes cancer cell migration and invasion. Many of these conclusions were based on experiments which employed treatment of cell lines representing different normal or cancer types with pan-CDK inhibitors (table 1), or employed overexpression of dominant-negative CDK5 constructs, or expression of anti-CDK5 siRNAs or shRNAs. It should be noted that CDK5 inhibitors used in these studies are not CDK5-specific, and they inhibit several other kinases [94,135–137] (table 1). Likewise, siRNAs and shRNA may have some off-target effects, while overexpression of dominant-negative CDK constructs is known to result in non-physiological effects. Hence, while the prospect of targeting CDK5 in various pathologies seems promising, conflicting results necessitate additional studies using approaches that allow to selectively inhibit CDK5 activity. It will also be important to use tissue-specific Cdk5 knockout mice in conjunction with mouse models of different

royalsocietypublishing.org/journal/rsob Open Biol. 10: 190287

pathological conditions (such as cancer) to conclusively resolve the role of CDK5 in physiology and in disease states.

Data accessibility. This article has no additional data.

Authors' contributions. S.S. and P.S. contributed equally in preparing this manuscript.

Competing interests. P.S. has served as a consultant at Novartis, Genovis, Guidepoint, The Planning Shop, ORIC Pharmaceuticals and Exo Therapeutics; his laboratory receives research funding from Novartis.

Funding. This work was supported by a grant no. R01 CA239660 (to P.S).

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
