## [Reviewer comments · Open Biology]

Review History

RSOB-19-0181.R0 (Original submission)

Review form: Reviewer 1 (Tej K. Pareek)

Recommendation

Accept with minor revision (please list in comments)

Do you have any ethical concerns with this paper?

Yes

Comments to the Author

The role of Cdk5 in pain signaling is missing from this comprehensive review

A detailed discussion (maybe a seperate section) on Cdk5 targeted drug efforts and its limitation and challenges will enhance the value of this article.

Review form: Reviewer 2

Recommendation

Accept with minor revision (please list in comments)

Do you have any ethical concerns with this paper?

No

Comments to the Author

This is a very interesting, comprehensive and well written review about the extra-neuronal functions of CDK5, with particular emphasis on its involvement in tumorigenesis. Figures are also well made and informative.

I have just two remarks:

- Authors never refer to the figures in the text. Please, insert the references to the figures in the text, where appropriate.
- The name of the kinase is sometimes capitalized while in other instances it is in lower case. Authors should make uniform the way of writing the kinase symbol, perhaps by always using uppercase characters (including chapter titles), unless they explicitly refer to an animal model ortholog.

Decision letter (RSOB-19-0181.R0)

07-Oct-2019

Dear Ms Sharma

We are pleased to inform you that your manuscript RSOB-19-0181 entitled "A kinase of many talents: non-neuronal functions of CDK5 in development and disease" has been accepted by the Editor for publication in Open Biology. The reviewer(s) have recommended publication, but also suggest some minor revisions to your manuscript. Therefore, we invite you to respond to the reviewer(s)' comments and revise your manuscript.

Please submit the revised version of your manuscript within 7 days. If you do not think you will be able to meet this date please let us know immediately and we can extend this deadline for you.

- 1) A text file of the manuscript (doc, txt, rtf or tex), including the references, tables (including captions) and figure captions. Please remove any tracked changes from the text before submission. PDF files are not an accepted format for the "Main Document".
- 2) A separate electronic file of each figure (tiff, EPS or print-quality PDF preferred). The format should be produced directly from original creation package, or original software format. Please note that PowerPoint files are not accepted.
- 3) Electronic supplementary material: this should be contained in a separate file from the main text and meet our ESM criteria (see <http://royalsocietypublishing.org/instructions-authors#question5>). All supplementary materials accompanying an accepted article will be treated as in their final form. They will be published alongside the paper on the journal website and posted on the online figshare repository. Files on figshare will be made available approximately one week before the accompanying article so that the supplementary material can be attributed a unique DOI.

Online supplementary material will also carry the title and description provided during submission, so please ensure these are accurate and informative. Note that the Royal Society will not edit or typeset supplementary material and it will be hosted as provided. Please ensure that the supplementary material includes the paper details (authors, title, journal name, article DOI). Your article DOI will be 10.1098/rsob.2016[*last 4 digits of e.g. 10.1098/rsob.20160049*].

- 4) A media summary: a short non-technical summary (up to 100 words) of the key findings/importance of your manuscript. Please try to write in simple English, avoid jargon, explain the importance of the topic, outline the main implications and describe why this topic is newsworthy.

Images

Data-Sharing

It is a condition of publication that data supporting your paper are made available. Data should be made available either in the electronic supplementary material or through an appropriate repository. Details of how to access data should be included in your paper. Please see <http://royalsocietypublishing.org/site/authors/policy.xhtml#question6> for more details.

Data accessibility section

Sincerely,
The Open Biology Team
mailto:openbiology@royalsociety.org

Reviewer(s)' Comments to Author:

Referee: 1

Comments to the Author(s)

The role of Cdk5 in pain signaling is missing from this comprehensive review
A detailed discussion (maybe a seperate section) on Cdk5 targeted drug efforts and its limitation and challenges will enhance the value of this article.

Referee: 2

Comments to the Author(s)

This is a very interesting, comprehensive and well written review about the extra-neuronal functions of CDK5, with particular emphasis on its involvement in tumorigenesis.
Figures are also well made and informative.

I have just two remarks:

- Authors never refer to the figures in the text. Please, insert the references to the figures in the text, where appropriate.
- The name of the kinase is sometimes capitalized while in other instances it is in lower case. Authors should make uniform the way of writing the kinase symbol, perhaps by always using uppercase characters (including chapter titles), unless they explicitly refer to an animal model ortholog.

Author's Response to Decision Letter for (RSOB-19-0181.R0)

See Appendix A.

Decision letter (RSOB-19-0287.R0)

02-Dec-2019

Dear Ms Sharma,

We are pleased to inform you that your manuscript entitled "A kinase of many talents: non-neuronal functions of CDK5 in development and disease" has been accepted by the Editor for publication in Open Biology.

Sincerely,

The Open Biology Team
mailto: openbiology@royalsociety.org

Appendix A

Peter Sicinski, M.D., Ph.D.

Professor of Genetics
Harvard Medical School

Department of Cancer Biology
Dana-Farber Cancer Institute

44 Binney Street
Boston, Massachusetts 02115
617.632.5005 tel, 617.632.5006 fax
peter_sicinski@dfci.harvard.edu

Dr. David Glover
Editor-in-Chief
Open Biology

November 28, 2019

Dear David,

Thank you for letting us know that our manuscript RSOB-19-0181 entitled "A kinase of many talents: non-neuronal functions of CDK5 in development and disease" has been accepted by the Editor for publication in Open Biology.

Our responses to Reviewers' suggestions are as follows:

Reviewer #1

The role of Cdk5 in pain signaling is missing from this comprehensive review

We believe that this topic is outside of the scope of our review, which focuses exclusively on non-neuronal functions of CDK5.

A detailed discussion (maybe a separate section) on Cdk5 targeted drug efforts and its limitation and challenges will enhance the value of this article.

In response, in the revised version we added a separate section on 'CDK5-specific' inhibitors.

Reviewer#2

Authors never refer to the figures in the text. Please, insert the references to the figures in the text, where appropriate.

This has been corrected.

The name of the kinase is sometimes capitalized while in other instances it is in lower case.

Authors should make uniform the way of writing the kinase symbol, perhaps by always using uppercase characters (including chapter titles), unless they explicitly refer to an animal model ortholog.

We revised the text accordingly.

We are now enclosing the revised version containing changes recommended by the Reviewers.

Thank you very much for your help.